# Evaluation of VIIRS Thermal Emissive Bands Long-Term Calibration Stability and Inter-Sensor Consistency Using Radiative Transfer Modeling

**Feng Zhang** [1,*], **Xi Shao** [1], **Changyong Cao** [2], **Yong Chen** [2], **Wenhui Wang** [1], **Tung-Chang Liu** [1]
**and Xin Jing** [1]

1  Cooperative Institute for Satellite Earth System Studies (CISESS), Earth System Science Interdisciplinary Center (ESSIC), University of Maryland, College Park, MD 20740, USA; xshao@umd.edu (X.S.); wenhui.wang@noaa.gov (W.W.); tcliu@umd.edu (T.-C.L.); xinjing@umd.edu (X.J.)
2  Center for Satellite Applications and Research (STAR), National Environmental Satellite, Data, and Information Service (NESDIS), National Oceanic and Atmospheric Administration (NOAA), College Park, MD 20740, USA; changyong.cao@noaa.gov (C.C.); yong.chen@noaa.gov (Y.C.)
*  Correspondence: zfsu@umd.edu

**Abstract:** This study investigates the long-term stability of the Suomi National Polar-orbiting Partnership (S-NPP) Visible Infrared Imaging Radiometer Suite (VIIRS) moderate-resolution Thermal Emissive Bands (M TEBs; M12–M16) covering a period from February 2012 to August 2020. It also assesses inter-sensor consistency of the VIIRS M TEBs among three satellites (S-NPP, NOAA-20, and NOAA-21) over eight months spanning from 18 March to 30 November 2023. The field of interest is limited to the ocean surface between 60°S and 60°N, specifically under clear-sky conditions. Taking radiative transfer modeling (RTM) as the transfer reference, we employed the Community Radiative Transfer Model (CRTM) to simulate VIIRS TEB brightness temperature (BTs), incorporating European Centre for Medium-range Weather Forecasts (ECMWF) reanalysis data as inputs. Our results reveal two key findings. Firstly, the reprocessed S-NPP VIIRS TEBs exhibit a robust long-term stability, as demonstrated through analyses of the observation minus background BT differences (O-B ΔBTs) between VIIRS measurements (O) and CRTM simulations (B). The drifts of the O-B BT differences are consistently less than 0.102 K/Decade across all S-NPP VIIRS M TEB bands. Notably, observations from VIIRS M14 and M16 stand out with drifts well within 0.04 K/Decade, reinforcing their exceptional reliability for climate change studies. Secondly, excellent inter-sensor consistency among these three VIIRS instruments is confirmed through the double-difference analysis method (O-O). This method relies on the O-B BT differences obtained from daily VIIRS operational data. The mean inter-VIIRS O-O BT differences remain within 0.08 K for all M TEBs, except for M13. Even in the case of M13, the O-O BT differences between NOAA-21 and NOAA-20/S-NPP have values of 0.312 K and 0.234 K, respectively, which are comparable to the 0.2 K difference observed in overlapping TEBs between VIIRS and MODIS. These disparities are primarily attributed to the significant differences in the Spectral Response Function (SRF) of NOAA-21 compared to NOAA-20 and S-NPP. It is also found that the remnant scene temperature dependence of NOAA-21 versus NOAA-20/S-NPP M13 O-O BT difference after accounting for SRF difference is ~0.0033 K/K, an order of magnitude smaller than the corresponding rates in the direct BT comparisons between NOAA-21 and NOAA-20/S-NPP. Our study confirms the versatility and effectiveness of the RTM-based TEB quality evaluation method in assessing long-term sensor stability and inter-sensor consistency. The double-difference approach effectively mitigates uncertainties and biases inherent to CRTM simulations, establishing a robust mechanism for assessing inter-sensor consistency. Moreover, for M12 operating as a shortwave infrared channel, it is found that the daytime O-B BT differences of S-NPP M12 exhibit greater seasonal variability compared to the nighttime data, which can be attributed to the idea that M12 radiance is affected by the reflected solar radiation during the daytime. Furthermore, in this study, we've also characterized the spatial distributions of inter-VIIRS BT differences, identifying variations among VIIRS M TEBs, as well as spatial discrepancies between the daytime and nighttime data.

**Keywords:** Visible Infrared Imaging Radiometer Suite (VIIRS); NOAA-21; NOAA-20; S-NPP; radiative transfer modeling; community radiative transfer model (CRTM); thermal emissive band (TEB); calibration; observation minus background difference (O-B); double-difference method

## 1. Introduction

The Visible Infrared Imaging Radiometer Suite (VIIRS) is a crucial instrument onboard the Suomi National Polar-orbiting Partnership (S-NPP) satellite, bridging the NASA Earth Observing System and the next generation Joint Polar Satellite System (JPSS) platforms since its launch in November 2011 [1]. Designed for seamless continuity with the legacy of Moderate Resolution Imaging spectroradiometer (MODIS), the S-NPP VIIRS has ushered in a new era of operational environmental remote sensing, contributing significantly to scientific research and applications in Earth's land, ocean, and atmosphere studies [2–4]. More than 20 VIIRS Environmental Data Records (EDRs) have been derived from its Sensor Data Records (SDRs), providing essential data on aerosols, cloud properties, fires, albedo, snow and ice, vegetation, sea surface temperature, ocean color, and nighttime visible light applications. Rigorous verification and validation efforts have been applied to S-NPP VIIRS SDR data [5], and numerous studies have examined the data consistency between S-NPP VIIRS and MODIS [6–9]. These studies have demonstrated that the VIIRS Thermal Emissive Bands (TEBs) closely align with similar bands of MODIS, exhibiting differences within a 0.2 K range, emphasizing the remarkable consistency between VIIRS and MODIS observations.

Following the launch of the S-NPP satellite, two additional JPSS satellites, namely NOAA-20 and NOAA-21, were successfully launched in November 2017 and November 2022, respectively. The three VIIRSs on these satellites will provide continuous Earth observations for over a decade. Detailed information on VIIRS TEB calibration algorithms, characteristics, and performance can be referenced in several prior studies [10,11]. The on-orbit VIIRS TEB calibration is performed on a scan-by-scan basis, employing a quadratic calibration algorithm. This algorithm utilizes observations from the V-grooved onboard calibrator blackbody (OBCBB) at a fixed scan-angle, with the space view (SV) providing the background offset for the calibration during each scan. Recent studies [12] have provided further insights into VIIRS TEB calibration.

Previous studies noted that significant differences exist in the spectral response functions (SRFs) of different instruments, such as VIIRS and MODIS. To mitigate the effects of the SRF differences, transfer references were employed, such as measurements from hyperspectral infrared sounders [7,13] like the Cross-track Infrared Sounder (CrIS) on SNPP and NOAA-20 spacecraft [5]. However, these intercomparison studies are limited in the number of spectral channels they can cover due to the mismatch between different instruments. Moreover, NOAA-21, NOAA-20 and S-NPP are on the same orbital plane and 25–50 min apart in orbital time. As a result, there are no nadir co-locations among these three VIIRS instruments. That is also one of the major reasons why we use radiative transfer modeling (RTM) as transfer reference to intercompare these three VIIRSs. In this study, following the methodology outlined by Liu et al. [14], the RTM has been applied as the transfer reference. The simulation of VIIRS TEB brightness temperatures (BTs) collocated with VIIRS observations is carried out using the Community Radiative Transfer Model (CRTM) [15]. This comprehensive evaluation covers all VIIRS M TEBs.

Two investigations with different time scales were conducted. The first study evaluates the long-term (2012–2020) stability of the NOAA STAR version 2 reprocessed S-NPP VIIRS moderate-resolution TEBs (M12–M16) data. The stability of VIIRS TEBs is crucial for the quality of downstream VIIRS environmental data record products, impacting variables like sea surface temperature and cloud products. The second study focused on analyzing the inter-sensor consistency of VIIRS TEB data among S-NPP, NOAA-20, and NOAA-21 in 2023. Notably, biases can emerge even with the same stable sensor, such as VIIRS, across

different satellites. Understanding and, where possible, resolving radiometric differences of this nature are necessary [5]. Ensuring consistency among NOAA-21, NOAA-20 and S-NPP is vital for extending existing data products and creating long-term global science datasets. Hence, the stability and consistency of VIIRS TEBs, the focal points of this study, are crucial for maintaining and upholding the data quality of downstream VIIRS EDR products and advancing Earth science research and climate applications.

In the following sections, we will first introduce the materials and methods for assessing VIIRS S-NPP TEB stability and inter-sensor VIIRS consistency in Section 2. Section 3 will present the results and analyses in detail. The discussion and conclusions are provided in Sections 4 and 5, respectively.

## 2. Materials and Methods

### 2.1. VIIRS Thermal Emissive Band Calibration and Characteristics

Among the 22 spectral bands of VIIRS, the five moderate-resolution TEBs (M TEBs: M12~M16, with a spatial resolution of 750 m at nadir) encompass a spectral region spanning from 3.6 to 12.5 μm. Table 1 lists the detailed VIIRS M TEB channel properties and primary applications.

**Table 1.** VIIRS moderate-resolution TEB channel properties and primary applications.

| VIIRS TEBs | Central Wavelength (μm) | | | Central Wavelength (μm) | Central Wavelength (μm) |
| --- | --- | --- | --- | --- | --- |
| | NOAA-21 | NOAA-21 | NOAA-21 | | |
| M12 | 3.688 | 3.696 | 3.693 | $H_2O$ | Sea surface temperature, Land surface type, Cloud mask. |
| M13 | 4.017 | 4.068 | 4.065 | − | Fires, Land surface type, Cloud mask, Dust. |
| M14 | 8.571 | 8.580 | 8.577 | $H_2O$ | Sea surface temperature, Land surface type, Cloud properties, Volcanic ash. |
| M15 | 10.640 | 10.693 | 10.710 | − | Sea surface temperature, Fires, VIIRS polar winds, Land surface temperature/type, Cloud properties, Cryosphere ice cover properties, Smoke/dust/volcanic ash. |
| M16 | 11.917 | 11.854 | 11.832 | $H_2O$ | Sea surface temperature, Fires, Land surface temperature/type, Cloud properties, Cryosphere ice cover properties, Volcanic ash. |

The VIIRS M TEB detectors are located on a cold focal plane assembly (FPA), nominally controlled at ~80 K with a passive radiative cooler. VIIRS TEBs rely on an onboard blackbody (BB) as the primary calibration source, in conjunction with the Space View ($SV$), which provides the instrument background reference. The calibration equation for the TEB, as detailed in the works of VIIRS SDR ATBD "https://ncc.nesdis.noaa.gov/documents/documentation/ATBD-VIIRS-RadiometricCal_20131212.pdf (accessed on 15 March 2023)", converts the back-ground-subtracted digital counts ($dn$) of each detector into spectral radiance entering the instrument aperture ($\underline{L_{ap}}$), which is averaged over the wavelength range covered by a spectral band. The $\underline{L_{ap}}$ is given by the following calibration equation:

$$\underline{L_{ap}} = \frac{F \sum_{i=0}^{2} c_i dn^i + (RVS_{SV} - RVS_\theta)\frac{\left[(1-\rho_{RTA})\underline{L_{RTA}} - \underline{L_{HAM}}\right]}{\rho_{RTA}}}{RVS_\theta}, \tag{1}$$

where $c_i$ is the calibration coefficients measured prelaunch, $RVS_\theta$ the response versus scan function ($RVS$) at the Earth View ($EV$) angle of incidence on the Half Angle Mirror ($HAM$), $\rho_{RTA}$ the reflectivity of the Rotating Telescope Assembly ($RTA$), and the $\underline{L_{RTA}}$ and $\underline{L_{HAM}}$ the averaged emitted radiances of the $RTA$ and $HAM$, respectively. $F$ represents the on-orbit degradation factor ($F$-factor), which is derived from every scan on-orbit and varies depending on the specific detector and the side of the $HAM$. The calibration of $EV$ scene radiance is adjusted to accommodate any changes in detector response by applying the $F$ factor, as

described in Equation (1). Continuous monitoring of detector response characteristics on orbit ensures the reliability of the calibration process outlined in Equation (1). For a more detailed understanding of the VIIRS TEB calibration algorithms, readers can refer to the following references: [10–12].

In this research, we employed radiative transfer modeling to assess the on-orbit stability and consistency of VIIRS M TEBs. This approach offers an independent evaluation of the onboard blackbody-based calibration stability and consistency, distinct from Equation (1).

Figure 1 illustrates the SRFs of the five VIIRS M TEBs, superimposed on the blackbody Planck Function curve at 292.5 K, corresponding to the BB temperature during VIIRS' normal operations. Notably, the spectral response of NOAA-21 M13 differs significantly from that of NOAA-20 and S-NPP VIIRS, as depicted in Figure 1b.

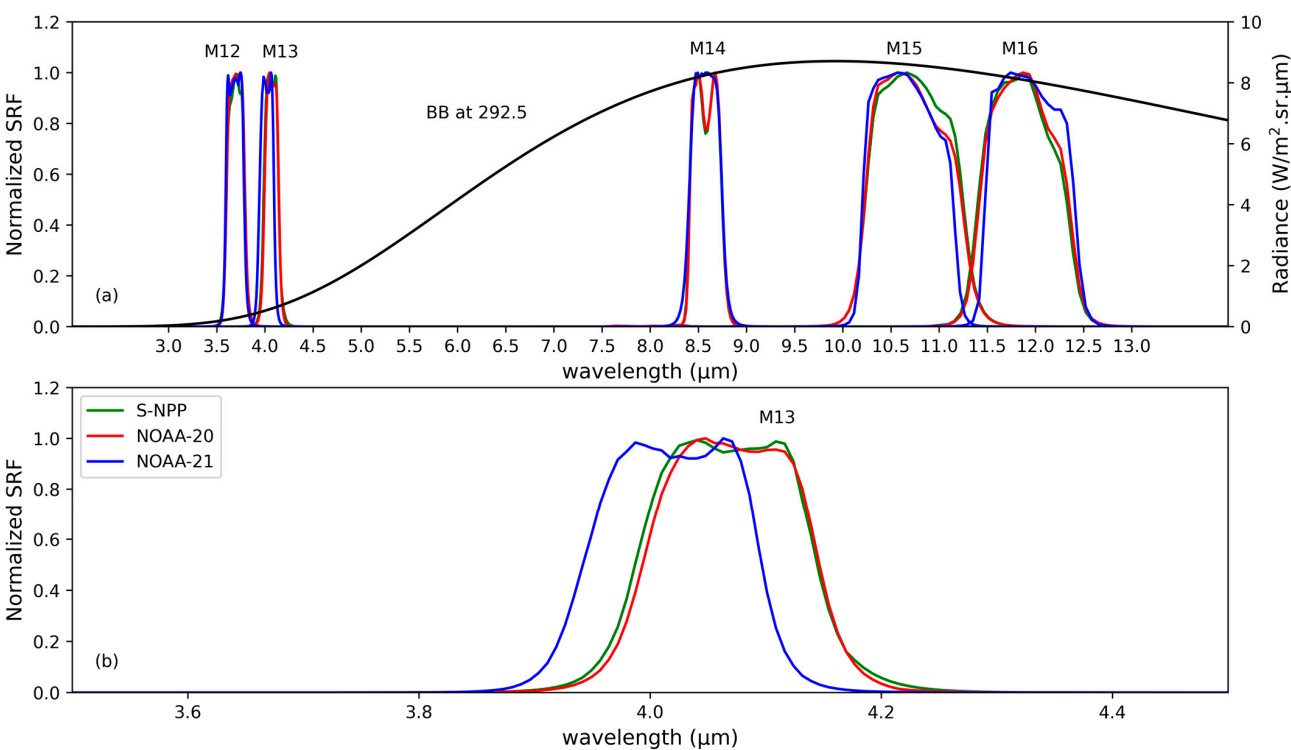

**Figure 1.** (**a**) VIIRS SRFs overlaid with blackbody Planck function curve at 292.5 K; (**b**) Zoomed VIIRS SRFs only for band 13 (M13).

Figure 2 illustrates the weighting function profiles (a) covering all four VIIRS TEBs except M13 and (b) specifically focusing on the M13 band. The weighting function is the derivative of transmittance ($\tau$) with respect to altitude (z), denoted as $d\tau/dz$. Here, it is derived from the 1976 US standard atmosphere model. The weighting function serves as an indicator of the predominant layer (at the surface or in the atmosphere) from which most of the radiation for a specific spectral band originates. All five M-bands (M12 to M16) clearly display their peak weighting function position at the surface level, indicating that they are primarily sensitive to surface conditions. Furthermore, except for the M13, all other bands exhibit minor differences in their weighting function profiles among S-NPP, NOAA-20, and NOAA-21. In the case of M13, NOAA-21 consistently shows a smaller $d\tau/dz$ across all altitudes compared to S-NPP and NOAA-20. This indicates that in the M13 band, NOAA-21 experiences fewer extinctions and larger transparency. Consequently, NOAA-21 is expected to observe a higher TOA reflected radiance, resulting in higher BTs for M13.

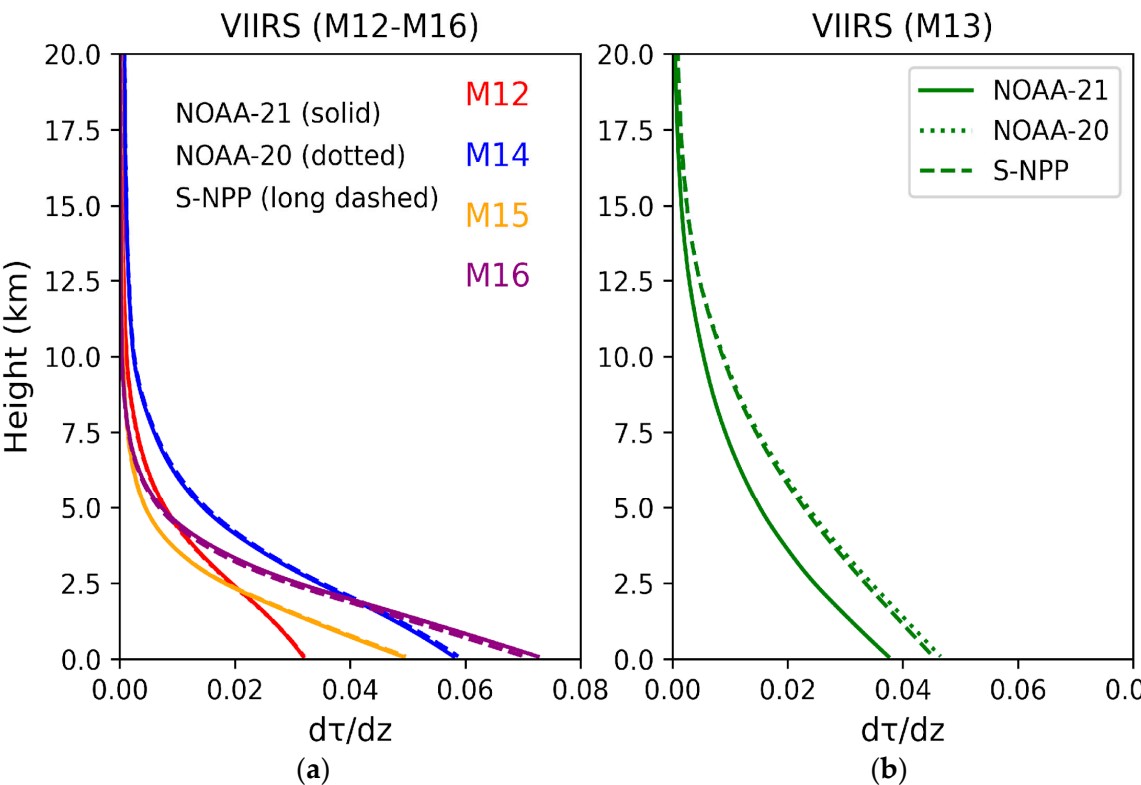

**Figure 2.** Weighting function profile of VIIRS TEB channels for S-NPP, NOAA-20, and NOAA-21, respectively. Where (**a**) shows all the channels except M13 and (**b**) shows only the M13 channel.

To further quantify the impact of the SRF differences among different VIIRS instruments, we conducted inter-comparisons of simulated Brightness Temperatures (BTs) relative to sea surface temperatures (SSTs), as illustrated in Figure 3. This analysis combined the CRTM with the 1976 US standard atmosphere model. In the CRTM simulation setup, the only disparity across different VIIRS instruments arose from the difference in SRFs. Figure 3 clearly shows that the differences between the BTs (ΔBTs) among different VIIRS instruments do not exceed 0.5 K for all TEBs, except for M13. For M13, the ΔBTs between NOAA-21 and NOAA-20/S-NPP significantly increase as SSTs rise. For example, when the SST is about 300 K, both the differences $BT_{NOAA-21} - BT_{NOAA-20}$ and $BT_{NOAA-21} - BT_{S-NPP}$ reach around 3.0 K. This substantial discrepancy is primarily due to the notable SRF differences between NOAA-21 and NOAA-20/S-NPP, as illustrated in Figure 1. This also infers that measurements of fires on land would register higher temperatures on NOAA-21 compared to NOAA-20 and S-NPP. Furthermore, given that VIIRS M12-M16 are all window channels with minimal or no water vapor absorption, adding humidity near ground levels will likely have negligible effects on simulated brightness temperature. Consequently, the findings depicted in Figure 3 and the conclusions drawn from them by using the US1976 atmosphere model remain unequivocally valid.

Above all, the establishment of a transfer reference that accounts for the spectral response differences among different VIIRS sensors is crucial for accurately assessing the inter-sensor consistency of VIIRS instruments.

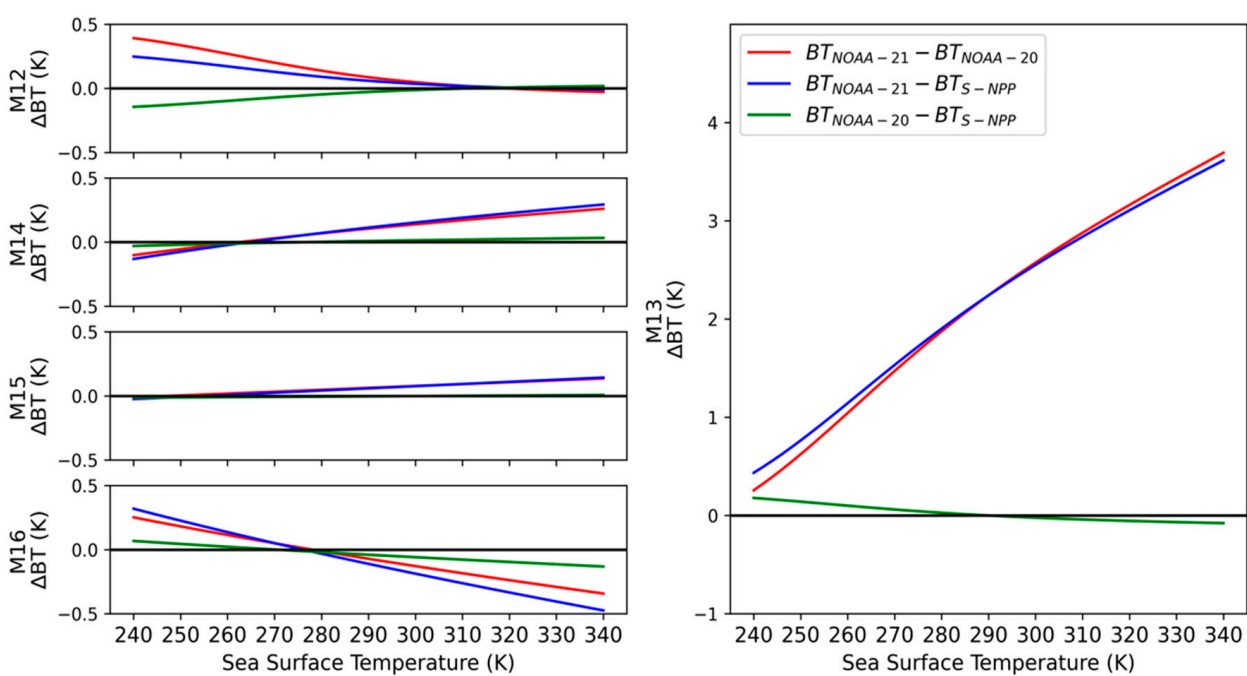

**Figure 3.** The differences of CRTM simulated BTs between different VIIRSs versus SSTs. The US1976 atmospheric profile has been used.

### 2.2. CRTM Radiative Transfer Modeling

Following the methodology established by Liu et al. [14], this work adopts radiative transfer modeling (RTM) as the transfer reference. In this study, the CRTM (version 2.3), specifically designed for simulating Top-of-Atmosphere (TOA) satellite-measured radiance, was utilized for simulating VIIRS TEB BTs, alongside collocated VIIRS observations. The model can be downloaded from the Joint Center for Satellite Data Assimilation (JCSDA) at "https://ftp.emc.ncep.noaa.gov/jcsda/CRTM/REL-2.3.0/ (accessed on 12 March 2023)". Quantitatively accounting for various factors, including Earth's surface reflection and radiation emission, single and multiple scattering, and gaseous absorption in the atmosphere, the CRTM also provides a comprehensive set of functions. These include forward modeling, adjoint modeling, tangent-linear modeling, and K-matrix modeling, catering to a wide range of modeling needs [15]. As a rapid sensor-channel-based RTM tool, the CRTM has found widespread applications in the calibration, assimilation, and various remote sensing endeavors, encompassing data from shortwave, infrared and microwave sensors [16,17].

Figure 4 provides a visual representation of the configuration for CRTM simulations in this research. ECMWF global reanalysis data, which were processed every 6 h with a spatial resolution of 0.25 degrees, served as essential inputs for CRTM modeling. These data include key parameters such as sea surface temperatures (SSTs), surface winds, and 37-level atmospheric profiles covering variables like water vapor, temperature, pressure, ozone, and more. The vertical profiles extended from the surface to about 1 hPa. To align ECMWF data with VIIRS observations, spatial and temporal interpolations were per-formed based on VIIRS observation time and locations. Additionally, the CRTM incorporated the Wu-Smith infrared water emissivity model for rough sea surfaces [18]. This scheme calculates ocean surface emissivity, considering the view angle, wavelength, and surface wind speed. Moreover, the CRTM incorporates the sensor-dependent spectral response function, thereby guaranteeing accurate model simulations aligned with observations.

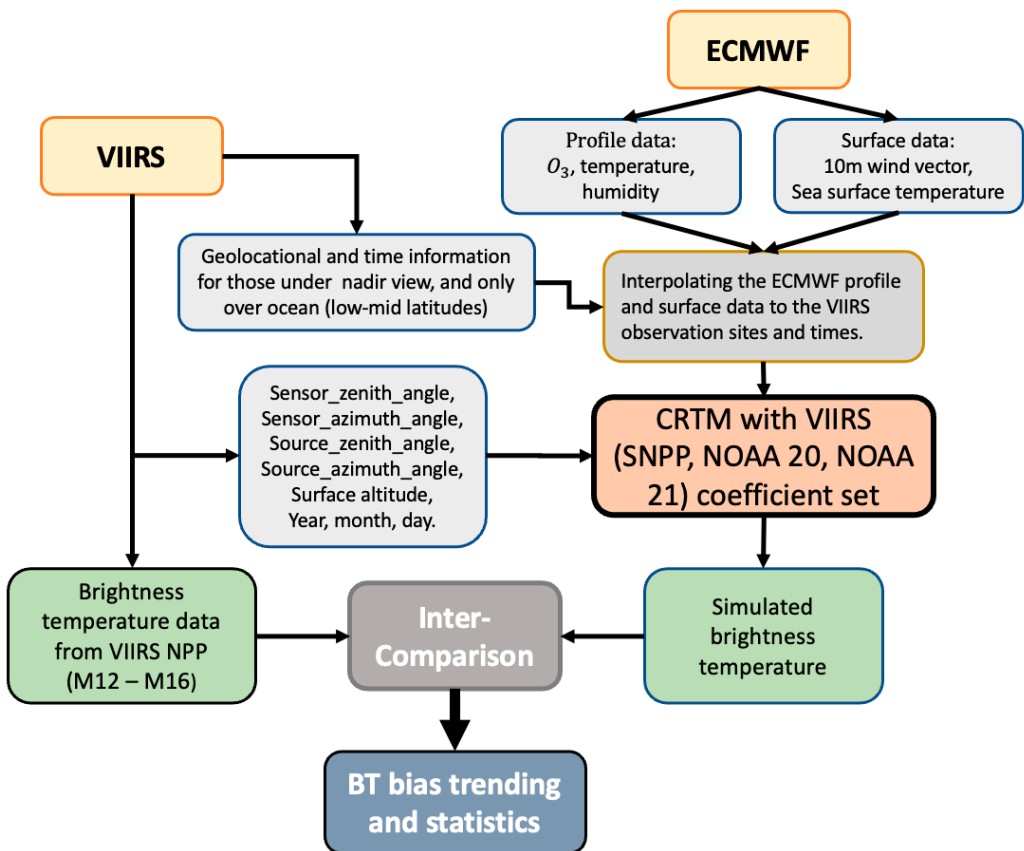

**Figure 4.** RTM simulation setup with CRTM as the simulator for long-term O-B difference evaluation of VIIRS TEBs.

In this study, the assessments of VIIRS TEB stability and inter-sensor consistency are based on the comparisons of VIIRS measurements (O) with their co-located CRTM simulations, i.e., observation minus background BT differences (O-B ΔBTs).

### 2.3. Scene Target Selection

This research mainly focuses on warm target regions situated over ocean within the low to mid-latitudes from 60°S to 60°N. Areas with solar zenith angles falling within the range of 80 to 100 degrees are excluded to avoid the terminator region, where the angle between the sun and the satellite's line of sight is relatively high.

For each O-B BT difference comparison, approximately 100 VIIRS data points around the center of the target location at satellite nadir were collected. Over these 100 points, the mean and standard deviation of VIIRS pixel BTs were recorded. Subsequently, those VIIRS measurements affected by scene non-uniformity (standard deviation of $BT \geq 0.3$ K) were removed. In addition, to mitigate the impact of cloud contamination, a straightforward yet effective cloud screen criterion was applied, removing any observations with absolute O-B BT difference equal to or exceeding 4 K across all VIIRS M TEBs. Further discussion regarding this cloud screening method will be presented in Section 3.2.3. After the uniformity and cloud contamination screening, the O-B dataset typically contained about 30,000 to 50,000 valid data points per day, ensuring the statistical robustness of the CRTM-based stability and consistency analysis. The ensemble of O-B BT differences over valid scenes during each day of interest was then utilized for comprehensive analyses, including mean O-B ΔBT calculations, mean double-difference between different O-B ΔBT (referred to as O-O ΔBT), uncertainty assessments, and trending analysis. Collectively, these steps contribute to the thorough evaluation of VIIRS TEB stability and inter-sensor consistency.

*2.4. Task Summary*

This study conducts two distinct investigations with different time frames and objectives. Details are as follows:

1.  Long-Term VIIRS stability evaluation (2012–2020):
    - Objective: To assess the long-term stability of the NOAA STAR version 2 reprocessed S-NPP VIIRS M TEB data [5].
    - Time Frame: February 2012 to August 2020.
    - Data Collection: the reprocessed S-NPP data "https://www.aev.class.noaa.gov/saa/products/search?sub_id=0&datatype_family=RPVIIRSSDR&submit.x=26&submit.y=12 (accessed on 1 April 2023)", ECMWF surface reanalysis data "https://cds.climate.copernicus.eu/cdsapp#!/dataset/reanalysis-era5-single-levels?tab=form (accessed on 1 April 2023)", and ECMWF pressure-level reanalysis data "https://cds.climate.copernicus.eu/cdsapp#!/dataset/reanalysis-era5-pressure-levels?tab=form (accessed on 1 April 2023)" were collected on the 15th day of each month during this timeframe.
    - Methodology: Monthly O-B ΔBT calculations were analyzed.

2.  Inter-VIIRS data consistency analysis (since 18 March 2023):
    - Objective: To analyze the inter-sensor consistency of M-TEB data across three VIIRS instruments: S-NPP, NOAA-20, and NOAA-21, each named after the satellite it is aboard.
    - Time Frame: from 18 March 2023 to 30 November 2023.
    - Data Collection: Daily operational data for S-NPP/NOAA-21/NOAA-20 "https://www.aev.class.noaa.gov/saa/products/search?sub_id=0&datatype_family=VIIRS_SDR&submit.x=22&submit.y=6 (accessed on 1 April 2023)", and 6-h ECMWF reanalysis surface and pressure-level data (The links are the same as before) were collected during this period.
    - Methodology: Daily calculations of both O-B ΔBTs and double-difference (O-O) ΔBTs were conducted. The double-difference analyses involve subtracting any pair of daily-mean O-B ΔBT values between S-NPP, NOAA-20, and NOAA-21 to derive inter-sensor VIIRS O-O ΔBTs.

The choice to conclude task 1 in August 2020 and set the endpoint for task 2 in November 2023 was primarily influenced by the availability of VIIRS TEB data when this research was conducted.

## 3. Results

*3.1. Long-Term Stability of VIIRS S-NPP M TEBs*

3.1.1. Analyses on the Long-Term Time Series

Figure 5 illustrates the long-term time series spanning from February 2012 to August 2020, displaying the O-B BT differences (O-B ΔBTs) between VIIRS observed (O) and CRTM modeled (B) for VIIRS S-NPP M-band TEBs (M12–M16). The figure also includes error bars representing uncertainties (blue vertical bar) and trend lines (red). The uncertainties are the standard deviation ($1\sigma$) of O-B ΔBTs on the 15th day of each month. The time series in Figure 5 reveals that the daily mean O-B ΔBTs for VIIRS M TEBs remain stable and consistent with ECMWF reanalysis data over the specified period. Except for M12, the mean O-B differences are all less than 0.26 K, with standard deviations $\leq$ 0.06 K, closely aligning with the 0.2 K BT difference between VIIRS and MODIS reported in prior studies [6–9]. For M12, the long-term mean O-B differences exhibit the largest values of 0.45 $\pm$ 0.1 K. Additionally, M12 also shows the highest uncertainty at 0.86 K, compared to M13 through M16 which have uncertainties of 0.53, 0.46, 0.53, and 0.57 K, respectively.

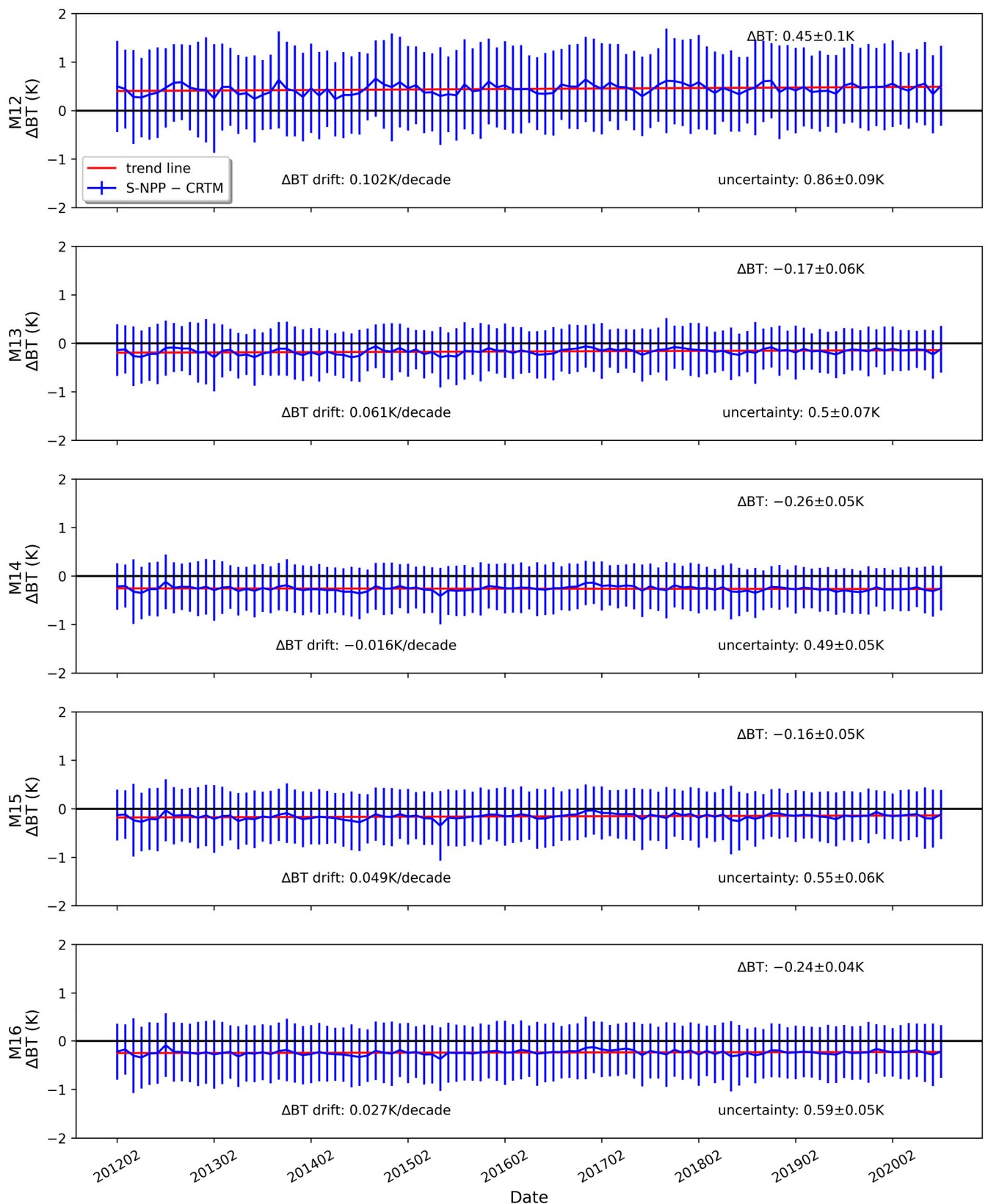

**Figure 5.** Long-term monthly trend of O-B ΔBT between VIIRS observed (O) and CRTM modeled (B) (blue) for VIIRS S-NPP M TEBs along with their uncertainties (1σ, the standard deviation of O-B ΔBTs on the 15th day of each month, bars) and trend lines (in red) from February 2012 to August 2020. The numbers shown here are the long-term means of monthly ΔBT with their standard deviations, along with those for uncertainties. The ΔBT drift for each band is also listed here.

To gain a deeper insight into the distinctive behavior of M12, we conducted a detailed analysis, specifically separating its O-B ΔBTs into daytime and nighttime. In this study, we classified daytime and nighttime measurements based on the solar zenith angle (SZA) as follows:

$$
\begin{aligned}
SZA < 80^o, & \quad daytime \\
SZA > 100^o, & \quad Nighttime
\end{aligned}
\tag{2}
$$

Figure 6 depicts the long-term O-B ΔBTs for M12 during daytime and nighttime, respectively. Particularly noteworthy is the observation that in M12, the long-term mean of O-B differences during daytime is significantly larger than that during nighttime, with values of 0.81 ± 0.16 K and 0.11 ± 0.08 K, respectively. Additionally, the long-term mean uncertainty during daytime for M12 reaches 0.91 K, also surpassing its nighttime uncertainty of 0.62 K. Clearly, the unique behavior of M12 primarily stems from its daytime measurements, which were contaminated by solar contributions. The M12 operates as a shortwave infrared channel. The solar contribution through sea surface reflection to TOA radiances is particularly notable during the daytime [16]. Consequently, the uncertainty stemming from the ocean bidirectional reflectance distribution function (BRDF) calculation results in a more pronounced CRTM simulation uncertainty for M12 over daytime than over nighttime. Hence, in the RTM-based TEB quality evaluation method, challenges arise from the intricacies and uncertainties inherent in the RTM simulation setup, including those associated with modeling surface emissivity and reflectivity. Therefore, it is expected that the daytime O-B BT differences of S-NPP M12 exhibit greater seasonal variability compared to the nighttime data, which is demonstrated by Figure 6.

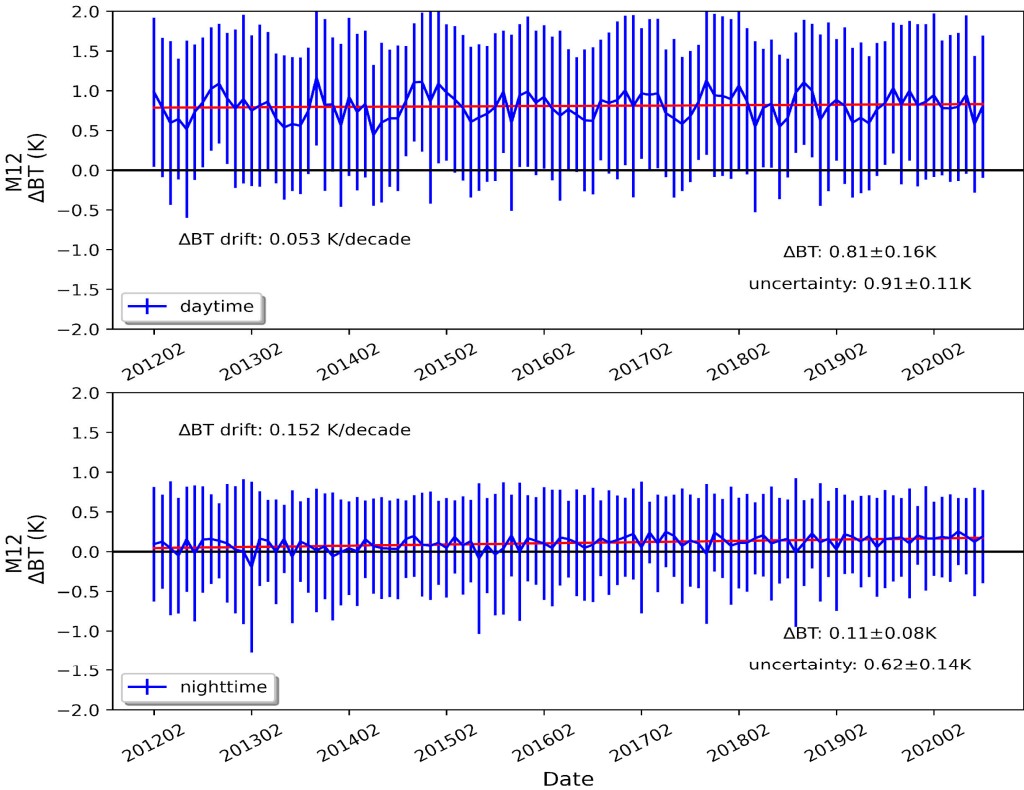

**Figure 6.** The same as Figure 5, except for M12 during daytime and nighttime, respectively. The trend lines are also shown in red.

### 3.1.2. Analysis on the Drifts of O-B BT Differences from 2012 to 2020

Next, our focus will shift to evaluating the long-term stability of VIIRS S-NPP TEBs through O-B difference analysis. The long-term time series of O-B BT differences are further analyzed using linear regression to calculate the decadal BT drift rate for each S-NPP

TEB, as indicated by the trend lines in Figures 5 and 6. The results of this analysis are concluded in Table 2, along with the corresponding 95% confidence interval (CI), providing a quantitative means to evaluate the stability of VIIRS S-NPP TEBs. Our results clearly show that VIIRS S-NPP TEBs exhibit radiometric stability, with the average decadal O-B ΔBT drift being less than 0.061 K/Decade for M13–M16, and slightly higher at 0.102 K/Decade for M12. As shown in Figure 6, for M12, the yearly drifts for daytime and nighttime are 0.053 K/Decade and 0.152 K/Decade, respectively. This highlights that the higher M12 yearly drift shown in Table 2 primarily stems from its nighttime data. This aspect deserves our attention in future VIIRS calibrations.

**Table 2.** Averaged yearly BT drifts ± 95% CI (K/Decade) of VIIRS NPP TEBs (M12–M16) derived from O-B analysis with CRTM simulations. ΔBT is the O-B BT difference, where O represents the VIIRS NPP observed BTs, and B represents the CRTM simulated BTs.

| VIIRS TEBs | Central Wavelength (μm) | Averaged Yearly BT Drift ± 95% CI (K/Decade) |
|:---:|:---:|:---:|
| M12 | 3.693 | 0.102 ± 0.076 |
| M13 | 4.065 | 0.061 ± 0.043 |
| M14 | 8.577 | −0.016 ± 0.037 |
| M15 | 10.710 | 0.049 ± 0.040 |
| M16 | 11.832 | 0.028 ± 0.035 |

Cao et al. (2021) conducted a comparative analysis of S-NPP VIIRS BTs and those acquired from the co-located Cross-track Infrared Sounder (CrIS) [5]. Their investigation revealed trends in the VIIRS-CrIS BT difference of about −0.03, −0.02, and −0.02 K/Decade for M13, and M15–M16, respectively. Here, utilizing the CRTM modeling as the transfer, we obtained slightly higher corresponding drifts for these three TEBs, with values of about 0.061, 0.049, and 0.028 K/Decade, respectively. However, the consistent discovery of similar magnitudes on the order of $10^{-2}$ K/Decade aligns with the findings of Cao et al. [5]. In addition, the RTM method is applied across all M TEBs (M12–M16) without imposing any spectral constraints. In contrast, Cao et al. [5] only evaluated three M bands (M13, M15, M16) comparing VIIRS and CrIS data. Undoubtedly, further information about the remaining two bands has been unveiled through using the RTM method.

Zou et al. [19] demonstrated the high radiometric stability performance of U.S. satellite microwave sounders, noting a trend within 0.04 K/Decade in the measured atmospheric temperature as indicative of reliable climate change detection. Our analyses, in Table 2, reveal drifts of about 0.016 and 0.028 K/Decade for S-NPP VIIRS M14 and M16, respectively, below 0.04 K/Decade. This demonstrates the reliability of observations from M14 and M16, making them suitable for climate change studies. While M15 exhibits a trend of about 0.049 K/Decade, it may still be considered suitable for such studies.

Our independent analysis reaffirms the overall stability of VIIRS S-NPP TEBs over ocean surfaces, particularly warm targets.

### 3.1.3. Analysis on O-B BT Differences against Scene Temperature

Figure 7 portrays O-B BT differences against scene temperature, offering valuable insights into the behavior of the S-NPP VIIRS TEBs. In this figure, the shaded area depicts 2-dimension O-B ΔBT distribution densities in unit $K^{-2}$, derived using O-B BT differences and S-NPP VIIRS BT observations from February 2012 to August 2020. About 61.0% to 76.9% of the scene BTs fall within the range of 290 K and above, emphasizing the prevalence of warm scene targets in this study. In Figure 7, a significant concentration of distribution density is revealed within the scene temperature range of 290 to 300 K for each M TEB. In this range, about 67%, 89%, 84%, 83% and 77% of ΔBTs have values no more than 0.6 K for M12–M16, respectively. Further examining the mean O-B ΔBTs versus scene temperature in Figure 7 demonstrates no significant scene temperature dependencies for the O-B differences of M14 to M16. However, for M13, the mean O-B BT bias curve comprises a

low-temperature segment marked by insignificant temperature dependence and a high-temperature segment (above 300 K) characterized by a downward trend with increased scene temperature. Only about 1% of BTs for M13 are above 300 K. Therefore, the downward trend in the high-temperature segment for M13 may be caused by data under-sampling. For M12, we've found a noticeable overall upward trend in the mean O-B ΔBT curve as the scene temperature increases, especially below 302 K. Above 302 K, where less than 1% of BTs fall within this range, the unusual performance of M12, characterized by a downward and then upward trend, suggests a potential association with data under-sampling.

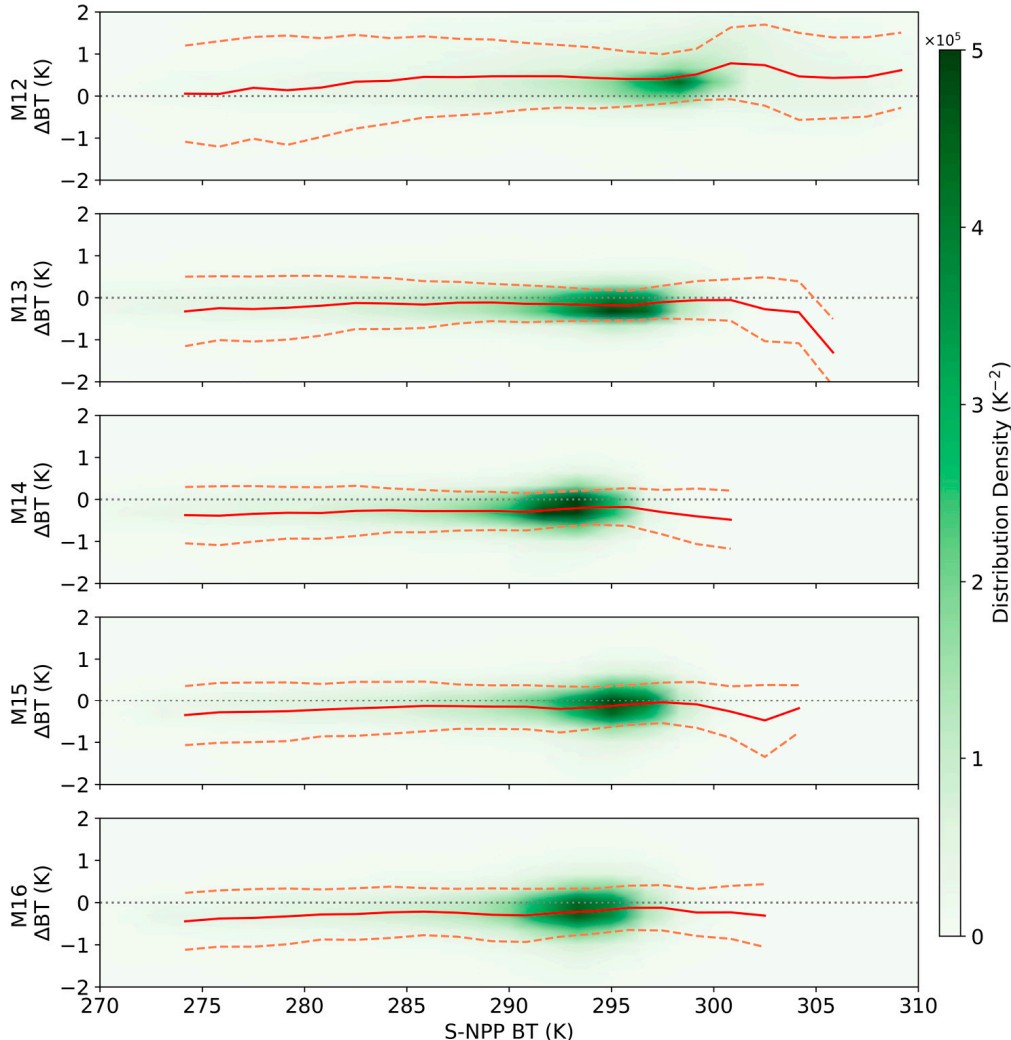

**Figure 7.** The dependence and distribution of O-B BT difference with respect to the BT measurement of S-NPP for M12-M16 TEBs. The O-B distribution statistics are derived using O-B BT difference and S-NPP VIIRS observations from 2012 to 2020. The curves overlayed with the distribution density plot show the mean (red solid) and one standard deviation from the mean (red dashed) of O-B BT differences in each BT bin.

To better understand the distinct behavior of M12 when scene temperature is below 302 K—higher scene temperatures correspond to higher ΔBTs—we further investigated the disparities between its daytime and nighttime observations. Figure 8, a counterpart to Figure 7, narrows its focus exclusively on S-NPP VIIRS M12 with a deliberate separation of daytime and nighttime data. The findings from Figure 8 reveal that during the day, ΔBTs for M12 are consistently large and stable, around 1 K across the whole BT range from 270 to 302 K. This is primarily attributed to solar contributions, as discussed above. Conversely, during the night, a notable upward trend appears with increasing BTs, characterized by

negative ΔBTs at lower temperatures (<about 287 K) and positive ΔBTs at higher temperatures (>about 287 K). Hence, the distinctive behavior of the M12 mainly results from the combined influence of daytime and nighttime measurements. These observations enhance our understanding of the temperature dependence of O-B differences within the context of S-NPP VIIRS TEBs, emphasizing the diurnal variations of O-B ΔBTs.

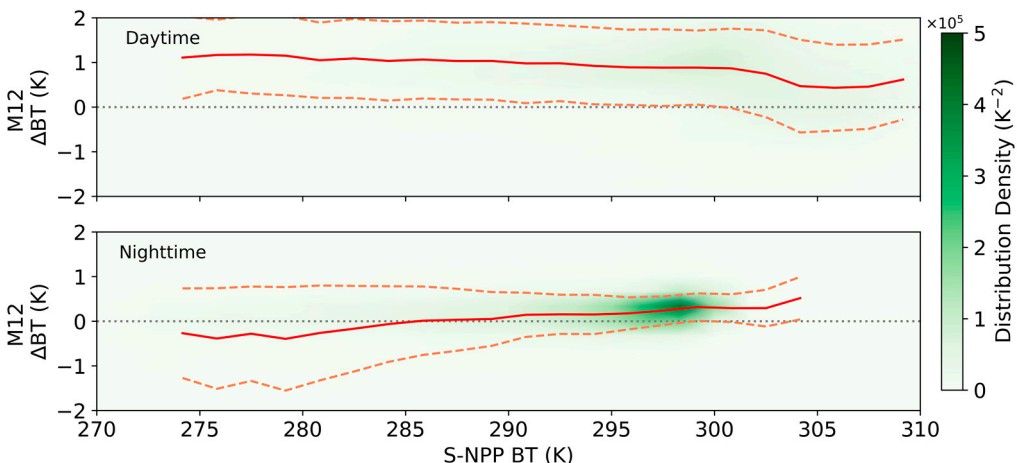

**Figure 8.** The same as Figure 7, but exclusively on the S-NPP VIIRS M12 with daytime and nighttime data separated.

### 3.2. Inter-Sensor Consistency of VIIRS M TEBs

As of February 2023, three VIIRS instruments onboard S-NPP, NOAA-20, and NOAA-21 have been operating steadily. Considering there were various events before March 17, such as mid-mission outgassing (MMOG) from 23 February to 25 February, OBC BB Warm-Up/Cool-Downs (WUCDs) from 10 March to 13 March and from 16 March to 18 March, which affected the quality of VIIRS TEBs' observations, we limited our analysis to data collected after 17 March to calculate the means of BT differences, their standard deviations and other variables.

#### 3.2.1. Analyses of the Time Series from 18 March to 30 November 2023

In Figure 9, the daily trends of O-B BT differences in 2023 are presented for VIIRS TEBs on NOAA-21, NOAA-20 and S-NPP. The gaps in data on 24 February, 12 March, 13 March and 4 May indicate instances where data were missing due to VIIRS TEB post-launch calibration events mentioned above. As shown in Figure 9, across all the M TEBs and the three VIIRS instruments on different satellites, the averaged O-B differences consistently stay below 0.46 K, with standard deviations not larger than approximately 0.09 K. These values are comparable to those present in the long-term monthly trends of SNPP VIIRS O-B ΔBTs shown in Figure 5. When disregarding the signs, M12 consistently displays the largest O-B ΔBTs, near 0.43 K, among all M TEBs for all VIIRS instruments, predominantly due to the shortwave solar contributions. While for M14 to M16, they exhibit similar small O-B ΔBTs with means not exceeding about 0.32, 0.17, and 0.24 K, respectively. In the case of M13, the mean O-B ΔBT for NOAA-21 is only 0.04 K, while for NOAA-20 and S-NPP, it is −0.27 K and −0.20 K, respectively. The significant discrepancy between NOAA-21 and NOAA-20/S-NPP in M13 is attributed to the substantial differences in their SRFs, as illustrated in Figure 1b. Certainly, even with CRTM as the reference, the non-linear atmospheric absorption effects resulting from variations in the SRF cannot be entirely removed.

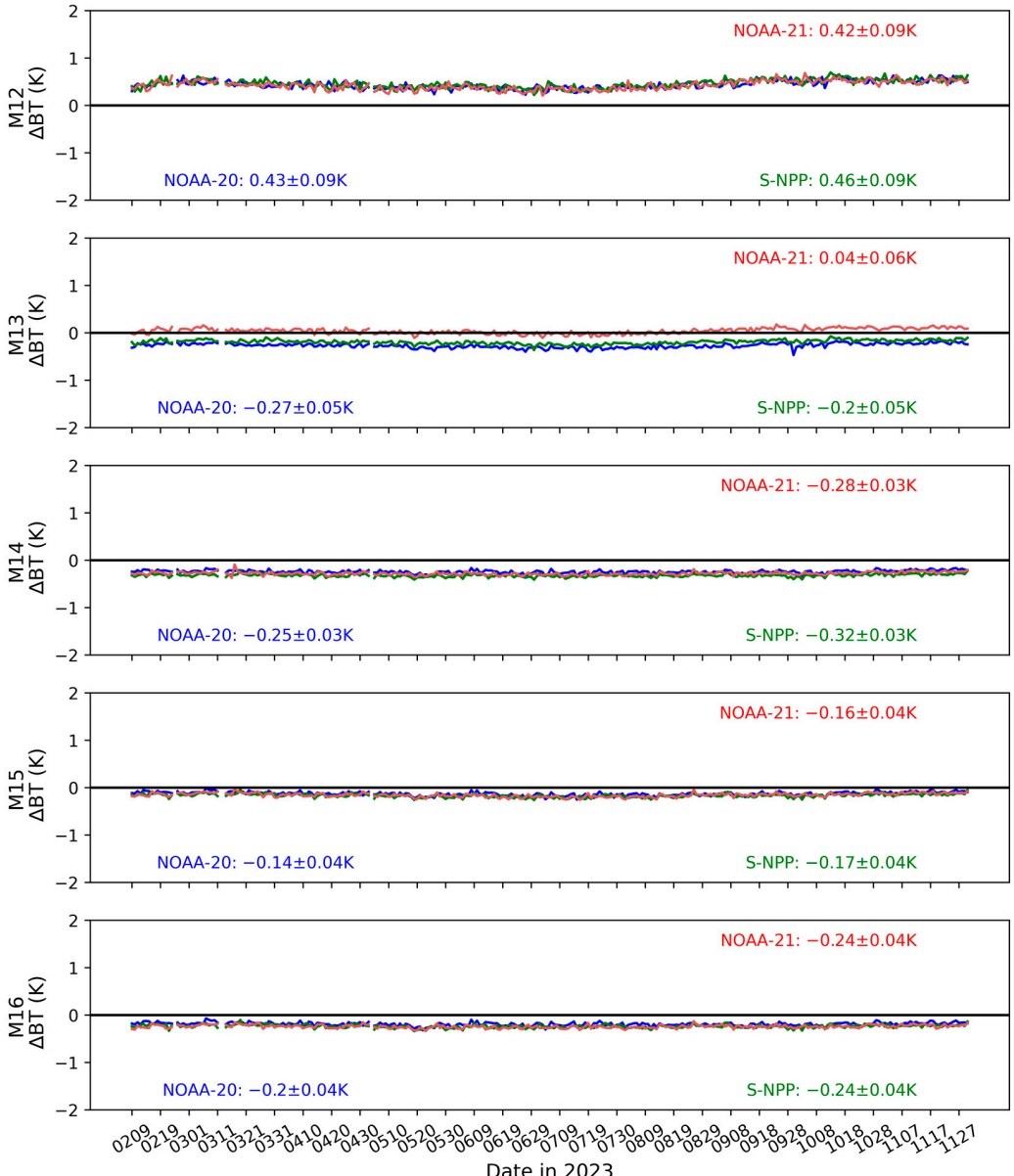

**Figure 9.** Daily trends of O-B ΔBTs for VIIRS S-NPP (green), NOAA-20 (blue), and NOAA-21 (red) TEBs (M12–M16). The numbers shown in each band are temporal means of daily mean ΔBT ± their standard deviations.

In addition, uncertainties in O-B ΔBTs, primarily stemming from the complexities and uncertainties in the RTM simulation setup (discussed previously, not shown here), are similar among the three VIIRSs for M12, M14, M15, and M16, with temporally averaged values of 0.81, 0.47, 0.53, and 0.57 K, respectively. However, for M13, noticeable differences in uncertainties are found among S-NPP, NOAA-20 and NOAA-21, with temporally averaged values of 0.46, 0.46, and 0.53 K, respectively. This is primarily attributed to the nonlinear effects resulting from the significant SRF differences between NOAA-21 and S-NPP/NOAA-20.

To better evaluate the inter-sensor consistency among different VIIRS instruments, we further conducted double-difference analyses by subtracting any pair of daily-mean O-B ΔBT values between S-NPP, NOAA-20, and NOAA-21 to derive inter-sensor VIIRS O-O ΔBTs. The double-difference technique for spaceborne instrument evaluation has been shown to be robust in previous studies [20,21]. Figure 10 shows the daily means of O-O ΔBTs between different pairs of these three VIIRS instruments. The numbers displayed in

each band represent the temporal means of daily O-O ΔBT ± standard deviations. Our findings reveal that, for all moderate-resolution VIIRS TEBs (excluding M13), the means of inter-VIIRS BT differences consistently remain below 0.08 K. These values are 1–2 orders of magnitude smaller than those of O-B ΔBTs shown in Figure 9, even for M12. Specifically, for NOAA-21–NOAA-20 and NOAA-20–S-NPP, their O-O ΔBTs in M12 have values of 0.013 K and 0.024 K, respectively, which are the smallest among all M TEBs, while the NOAA-21–NOAA-20 value of −0.038 K for M12 falls within the range of the other bands. This demonstrates that the double-difference approach mitigated uncertainties and biases inherent to CRTM simulations, including those originating from solar contributions in M12, establishing a robust mechanism for assessing inter-sensor VIIRS consistency. For M13, the SRF of NOAA-21 significantly differs from those of NOAA-20 and S-NPP. That leads to inter-VIIRS BT differences between NOAA-21 and NOAA-20/S-NPP are about 0.312 and 0.234 K, respectively, which are 1–2 orders of magnitude larger than those in the other bands, but still comparable to the 0.2 K difference between VIIRS and MODIS.

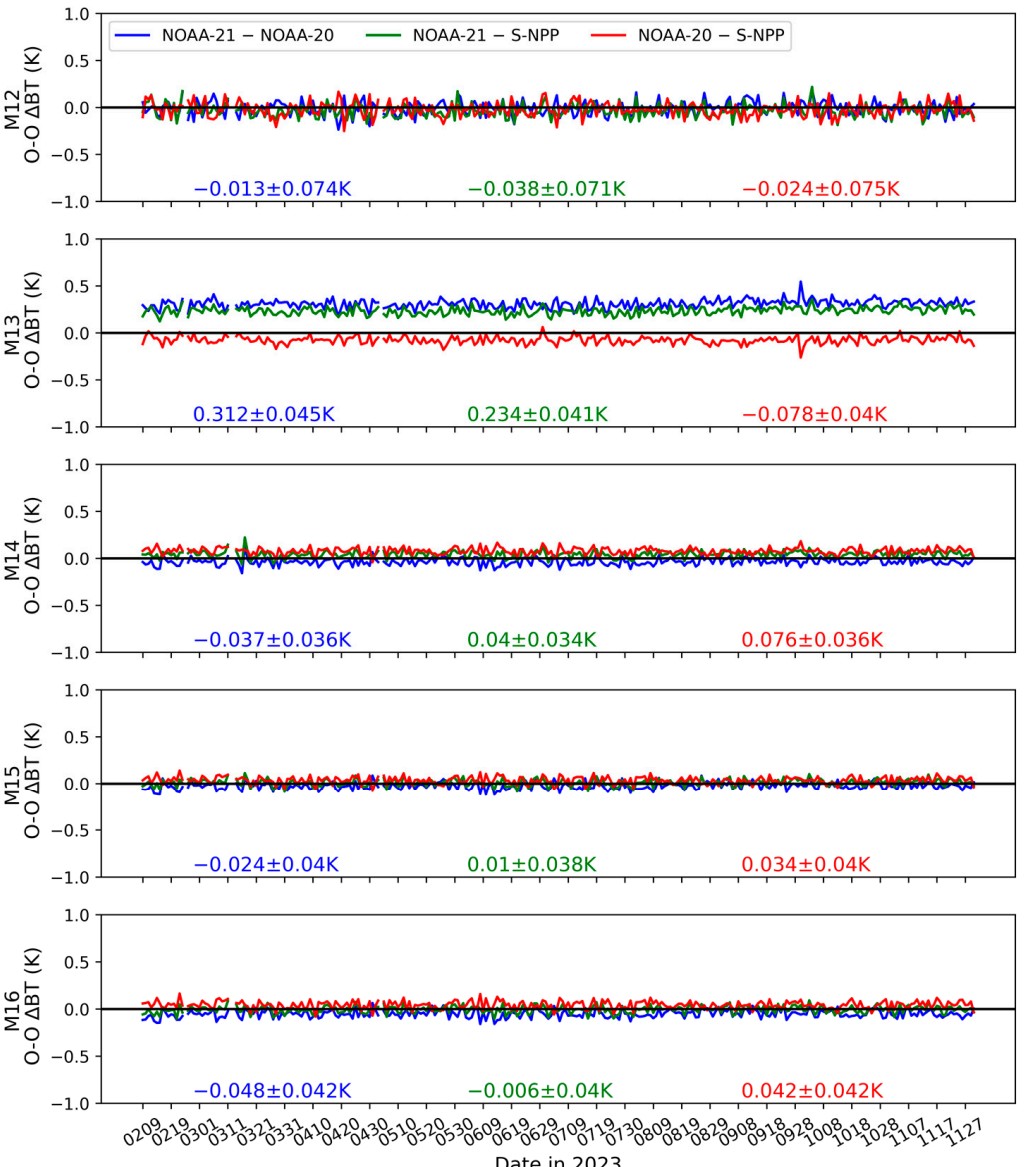

**Figure 10.** Comparisons of daily-mean O-O ΔBTs between different pairs among S-NPP/NOAA-20/NOAA-21. The numbers shown in each band are temporal means of O-O ΔBT ± standard deviations.

Figure 11 directly compares the temporal means of O-O ΔBTs (K) among these three VIIRS instruments for each M TEB in bar plots. Clearly, the O-O ΔBTs among different VIIRS instruments exhibit different performances for different bands. For example, for M15 and M16, the averaged O-O ΔBTs are smallest between NOAA-21 and S-NPP, while in M12 and M14, the best consistency can be found between NOAA-21 and NOAA-20.

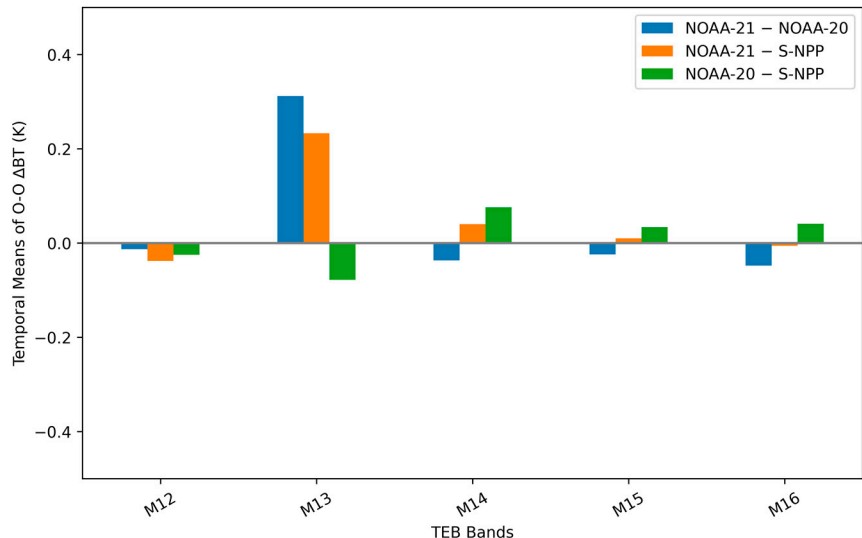

**Figure 11.** Comparisons of the temporal means of O-O ΔBTs (K) among S-NPP/NOAA-20/NOAA-21 in bar plots.

Furthermore, for better comparisons, all temporal averaged O-B and O-O ΔBTs along with their standard deviations for these three VIIRS instruments are also summarized in Tables 3 and 4.

**Table 3.** Temporal mean of daily averaged O-B ΔBTs (K) along with their standard deviations for NOAA-21, NOAA-20, and S-NPP VIIRS TEBs.

| VIIRS TEBs | O-B $\underline{\Delta BT} \pm \sigma$ (K) ($\underline{\Delta BT}$: Temporal Mean of Daily Mean ΔBT; $\sigma$: Standard Deviation) | | |
|---|---|---|---|
| | **NOAA-21** | **NOAA-20** | **S-NPP** |
| M12 | $0.42 \pm 0.09$ | $0.43 \pm 0.09$ | $0.46 \pm 0.09$ |
| M13 | $0.04 \pm 0.06$ | $-0.27 \pm 0.05$ | $-0.20 \pm 0.05$ |
| M14 | $-0.28 \pm 0.03$ | $-0.25 \pm 0.03$ | $-0.32 \pm 0.03$ |
| M15 | $-0.16 \pm 0.04$ | $-0.14 \pm 0.04$ | $-0.17 \pm 0.04$ |
| M16 | $-0.24 \pm 0.04$ | $-0.20 \pm 0.04$ | $-0.24 \pm 0.04$ |

**Table 4.** Temporal mean of daily averaged O-O ΔBTs (K) along with their standard deviations for NOAA-21, NOAA-20, and S-NPP VIIRS TEBs.

| VIIRS TEBs | O-O $\underline{\Delta BT} \pm \sigma$ (K) ($\underline{\Delta BT}$: Temporal Mean of Daily Mean ΔBT; $\sigma$: Standard Deviation) | | |
|---|---|---|---|
| | **NOAA-21–NOAA-20** | **NOAA-21–S-NPP** | **NOAA-20–S-NPP** |
| M12 | $-0.013 \pm 0.074$ | $-0.038 \pm 0.071$ | $-0.024 \pm 0.075$ |
| M13 | $0.312 \pm 0.045$ | $0.234 \pm 0.041$ | $-0.078 \pm 0.040$ |
| M14 | $-0.037 \pm 0.036$ | $0.040 \pm 0.034$ | $0.076 \pm 0.036$ |
| M15 | $-0.024 \pm 0.040$ | $0.01 \pm 0.038$ | $0.034 \pm 0.040$ |
| M16 | $-0.048 \pm 0.042$ | $-0.006 \pm 0.040$ | $0.042 \pm 0.042$ |

3.2.2. Analyses on the Relationship between ΔBTs and Scene Temperatures

As we know, VIIRS calibration biases usually depend on scene temperatures. Therefore, Figure 12 presents (a) the relationship between O-B ΔBTs and sea surface temperatures (SSTs) and (b) the relationship between O-O ΔBTs and SSTs. The SST covers a range from 272 to 305 K, divided into 33 bins, each with a width of 1 K. Since this study focuses on warm temperature targets, this SST range effectively covers most of the cases. From this figure, we observe clear increases in O-B ΔBTs as SSTs rise, especially when SST is below 300 K across all M TEBs. However, no apparent dependencies of O-O ΔBT on SST are observed for all TEBs except for M13. For these TEBs, the O-O ΔBTs consistently hover close to zero, significantly smaller than the corresponding O-B ΔBTs. Evidently, the double-difference method successfully alleviates the inherent uncertainties and biases associated with CRTM simulations. For M13, due to the SRF differences, the O-O ΔBTs between NOAA-21 and NOAA-20/S-NPP are around 0.2 K. This deviation is evident, deviating from zero, yet it remains comparable to the 0.2 K difference between VIIRS and MODIS.

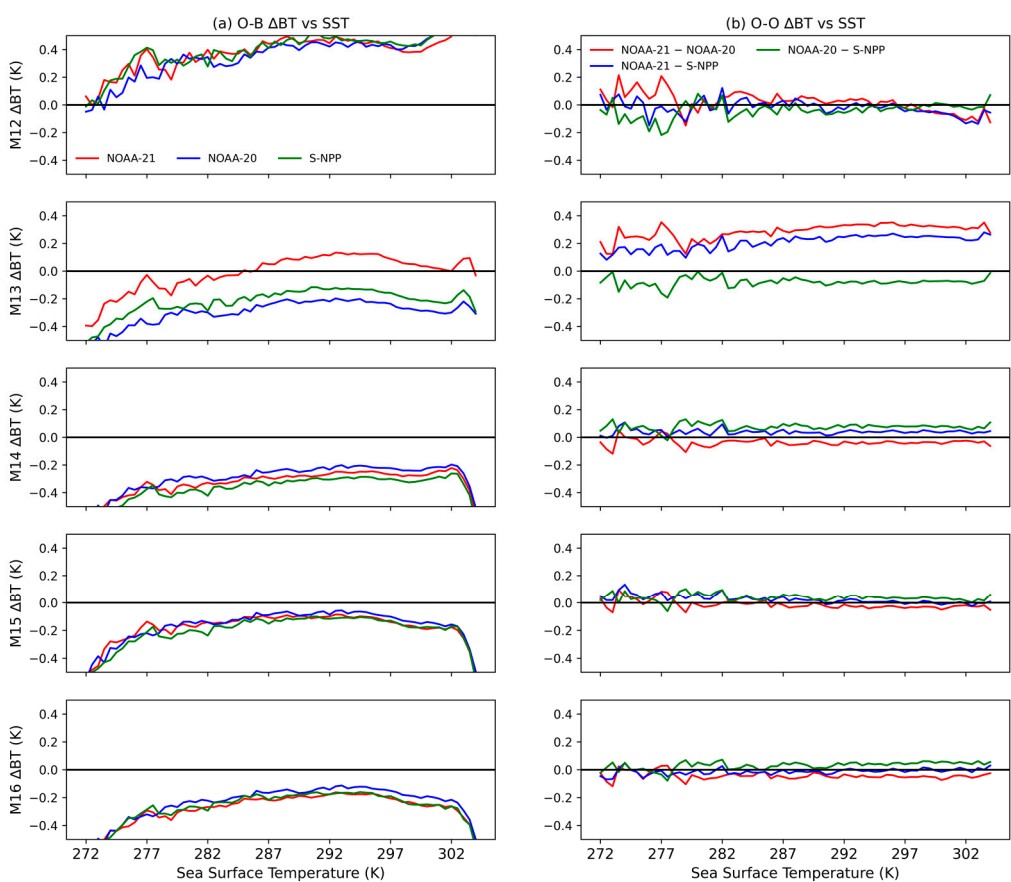

**Figure 12.** (**a**) The relationship between O-B ΔBTs and SSTs; (**b**) the relationship between O-O ΔBTs and SSTs. The SSTs covers a range from 272 to 305 K, which is divided into 33 bins, each with a width of 1 K.

Substantial differences in SRFs between NOAA-21 and NOAA-20/S-NPP, particularly in M13, can result in notably steep slopes, reaching about 0.035 K/K, in the changes of direct BT differences with SSTs rising, as shown in Figure 3. However, the double-difference method significantly mitigates this false large upward trend for M13. As shown in Figure 12, the slope of O-O ΔBTs between NOAA-21 and NOAA-20/S-NPP with rising SSTs consistently stays below 0.0033 K/K, an order of magnitude smaller than the corresponding rates in the direct BT comparisons. This highlights the effectiveness of the double-difference approach in minimizing the influence of SRF differences when assessing inter-sensor consistency.

### 3.2.3. Analyses on the Spatial Variation of O-O ΔBTs

In this study, we further analyzed the geographical distributions of the O-O ΔBTs, considering grids with resolution of 2° × 2°. The O-O ΔBT values were aggregated within each grid, and their respective grid-means are presented in Figure 13, showcasing comparisons of (a) NOAA-21–NOAA-20, (b) NOAA-21–S-NPP, and (c) NOAA-20–S-NPP, respectively. From this figure, the O-O grid-mean ΔBTs exhibit similar spatial distributions among (a), (b) and (c) in each TEB, except for M13. In (a) and (b) for M13, warm colors (red) dominate most ocean areas, especially within 40°S to 40°N. This indicates that NOAA-21 measurements consistently register higher BTs compared to both NOAA-20 and S-NPP, which is mainly attributed to the smallest weighting function of NOAA-21 (Figure 2). In addition, apart from M13, M12 consistently displays larger spatial variations than the other TEBs. This mainly results from solar contribution through the ocean reflection during the daytime, as discussed earlier. Moreover, in general, the grid-mean O-O ΔBTs are very small for all TEBs, typically falling within the range of −0.2 to 0.2 K. This demonstrates the effectiveness of our cloud screening method, wherein any observations with an absolute O-B BT difference equal to or exceeding 4 K across all VIIRS M TEBs have been removed. However, we still observe the presence of O-O ΔBTs over the Intertropical Convergence Zone (ITCZ) for M15 and M16, suggesting that our cloud screening method may not entirely remove certain types of clouds, such as cirrus clouds. Cirrus clouds are high-level clouds with semi-transparent characteristics, leading to O-B ΔBT values below 4 K. Therefore, further analyses, for example, adjustments to cloud screening thresholds, especially for cirrus, should be considered.

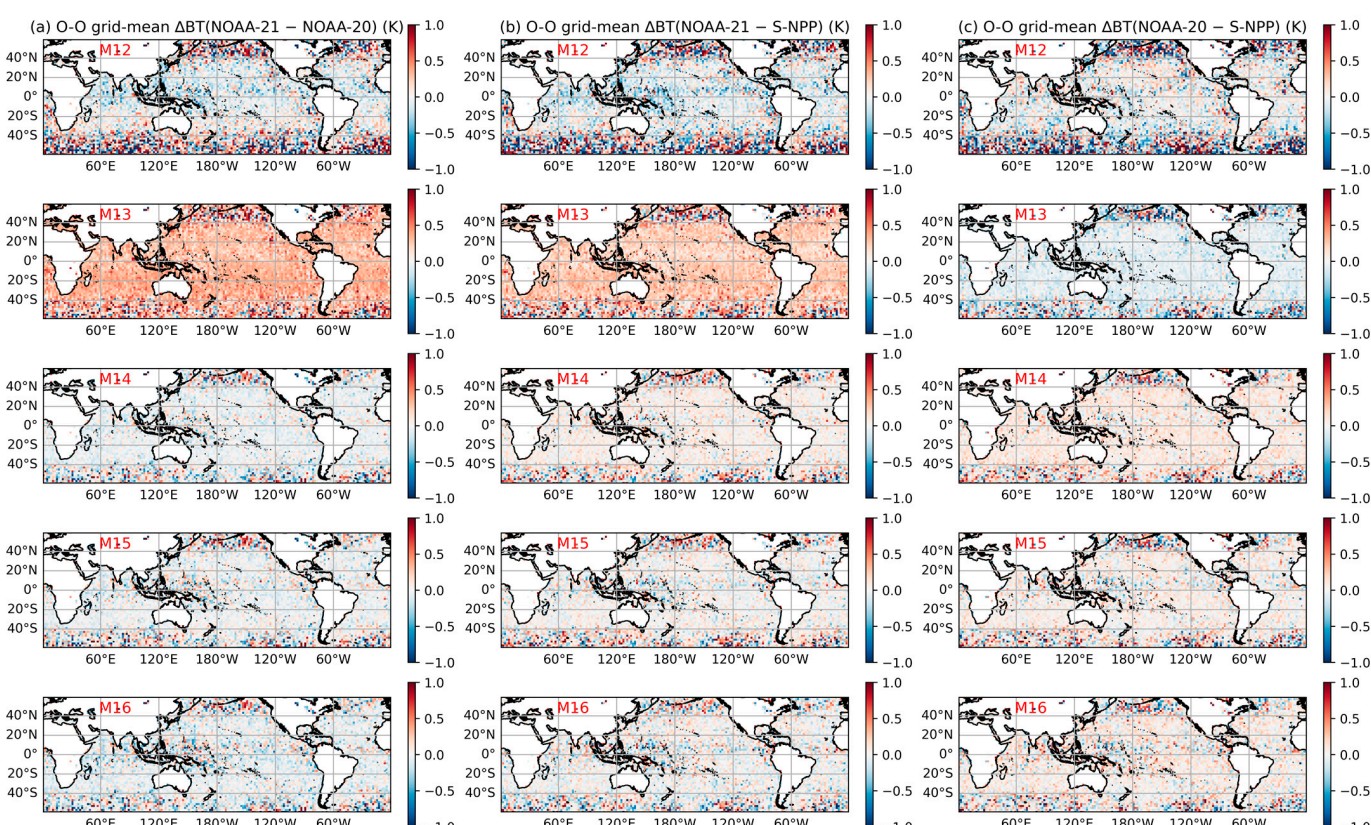

**Figure 13.** Longitude–latitude distributions of grid-mean O-O ΔBTs for (**a**) NOAA-21–NOAA-20, (**b**) NOAA-21–S-NPP, and (**c**) NOAA-20–S-NPP, respectively.

Figure 14 illustrates the longitude–latitude distributions of grid-mean O-O ΔBTs for (a) daytime and (b) nighttime specifically for those between NOAA-21 and NOAA-20. Except for M12, the spatial distributions of grid-mean O-O ΔBTs for all other M TEBs remain consistent, particularly evident between 40°S and 40°S, during both daytime and nighttime.

For M12, the contrast is evident as the variations of O-O ΔBTs during daytime (Figure 14a) are notably higher than those during nighttime (Figure 14b), offering evidence of solar contribution in M12 during daytime even after employing the double-difference method for mitigation. Furthermore, quite large values of O-O ΔBTs exist in high latitudes (>40°) during the night. For M12, this is also partially due to the solar contamination. In this study, the classification of daytime and nighttime measurements relies on the solar zenith angle (SZA) determined by Formula (2). Therefore, at high latitudes, solar contribution may still exist even when $SZA$ exceeds 100°. More discussions are in Section 4. Furthermore, similar variations in spatial distributions between daytime and nighttime data are observed for NOAA-21 versus S-NPP and NOAA-20 versus S-NPP for all TEBs (not shown).

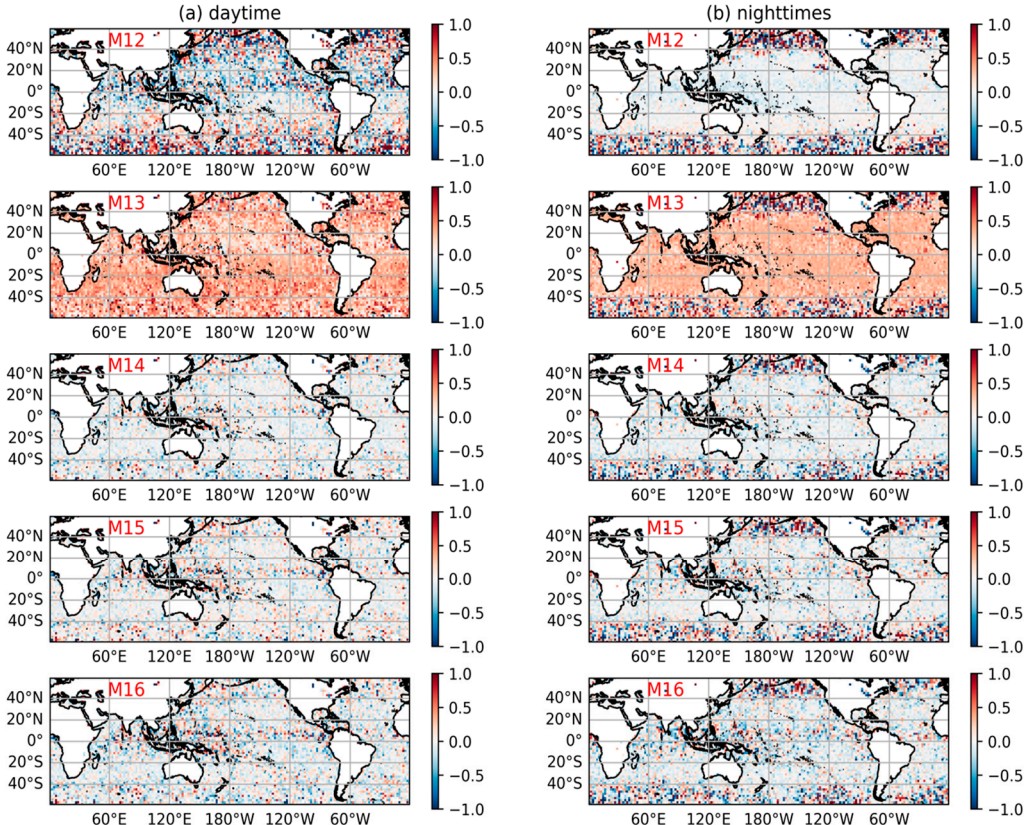

**Figure 14.** Longitude–latitude distributions of grid-mean O-O ΔBTs of NOAA-21–NOAA-20 for (**a**) daytime and (**b**) nighttime.

The RTM-based approach effectively demonstrates the excellent inter-sensor consistency of VIIRS TEBs, even in the M13 band. Using the CRTM simulation as the transfer reference, this study offers a comprehensive quality evaluation of VIIRS TEB data to ensure consistency among S-NPP, NOAA-20, and NOAA-21.

## 4. Discussion

Based on the results of this study, several discussion points arise as follows.

Firstly, significant differences in the SRFs among NOAA-21, NOAA-20, and S-NPP VIIRS instruments are noted for M13. While the double-difference method shows that these SRF differences have been accounted for to a considerable extent, it is essential to acknowledge that its non-linear effects cannot be entirely removed.

Secondly, at high latitudes in both the northern and southern hemispheres, there exist quite large absolute values of O-O ΔBTs during nighttime for all VIIRS TEBs (Figure 14). These high latitude regions are near the day–night terminator. There will be changing sunlight irradiating on the spacecraft and VIIRS instrument as the spacecraft transits between day and night until the spacecraft solar zenith angle is larger than 118.4° [22]. The

spacecraft and instrument temperature can experience large variation through thermal coupling during the transition over the day–night terminator region [23]. The blackbody calibration target temperature on VIIRS is monitored with six thermistors and the average temperature of these six thermistors is used as the representative temperature to characterize the blackbody calibration target for VIIRS TEB calibration. The non-uniformity of VIIRS blackbody temperature can cause the deviation of the calculated blackbody temperature from actual temperature observed by the VIIRS TEB detectors; such deviations are the most significant over the day–night terminator transition region and can cause calibration biases. Further research is needed to understand this high latitude O-O ΔBT biases by correlating the O-O TEB BT biases with onboard blackbody temperature nonuniformity and space view counts [23]. The above analysis is also applicable to the noticeable positive drift of approximately 0.152 K/decade in S-NPP M12 during the night. When we only take those with SZA > 118.4° as nighttime data instead of Formula (2), this drift largely reduced to about 0.1 K/decade with a ~33% decrease, indicating the solar contaminations in S-NPP nighttime data even when SZA is larger than 100°. However, the residual trend of about 0.1 K/decade still necessitates the inter-comparison with observations from other sensors such as MODIS [7,13].

Thirdly, while the cloud screening method employed in this study is effective, it may not remove all observations contaminated by clouds, such as those associated with cirrus clouds. As a result, further investigations, including adjusting cloud screening thresholds, should be conducted.

Fourthly, abnormal, or irregular satellite movements, such as jiggers, can introduce variability to the satellite's orientation and affect the geolocation accuracy. For VIIRs, its geolocation is maintained with onboard attitude determination and control system (ADCS) and on-orbit updates of geolocation parameters [24]. The VIIRS geolocation accuracy is continuously monitored, and its uncertainty is determined to be of subpixel level and about 100 m in either the along-scan or along-track direction [5,25]. The effects of geolocation uncertainty should be taken into consideration, particularly in areas with substantial spatial variation of surface emissivity. The co-location of CrIS and VIIRS on the same satellite suggests that they experience similar noises resulting from the same unexpected satellite movements and anomalies. As a result, while smaller BT differences are observed, anomalies arising from these movements cannot be addressed solely by comparing co-located VIIRS and CrIS data. This emphasizes the importance of introducing an independent method, such as the RTM modeling method, to evaluate the VIIRS performance alongside the co-located instrument comparisons.

## 5. Conclusions

This work employs radiative transfer modeling as the transfer reference. The Community Radiative Transfer Model (CRTM) is applied to simulate VIIRS TEB BTs using ECMWF reanalysis data as inputs for the collocated VIIRS observations. All analyses in this paper are confined to the clear-sky ocean surface between 60° S and 60° N as the areas of interest. Two significant investigations with different time scales were conducted. The first study evaluated the long-term (2012–2020) stability of S-NPP VIIRS TEBs using the NOAA STAR version 2 reprocessed S-NPP VIIRS moderate-resolution (M12–M16) TEBs data, based on the observation minus background BT differences between VIIRS measurements (O) and CRTM simulations (B). The second study focused on assessing the inter-sensor VIIRS operational TEB data consistency among S-NPP, NOAA-20, and NOAA-21 over eight months from 18 March 2023 to 30 November 2023, as revealed through the double-difference analysis method. This method is carried out by subtracting any pair of daily-mean O-B BT differences among the three satellites. In fact, NOAA-21, NOAA-20 and S-NPP are on the same orbital plane and are 25–50 min apart in orbital time. Hence, there are no nadir co-locations among these three VIIRS instruments. This is one of the major reasons why we use the ECMWF background model as a transfer reference to intercompare these three VIIRSs.

We found that, firstly, there is robust long-term stability observed in S-NPP VIIRS TEBs. The drifts of the O-B BT differences are consistently found to be less than 0.102 K/Decade for S-NPP VIIRS bands M12–M16. M14 and M16 measurements can be reliably utilized for climate change studies due to their drifts below 0.04 K/Decade. Secondly, excellent inter-sensor consistency is observed among different VIIRS instruments. For all moderate-resolution VIIRS TEBs (excluding the M13), the means of inter-VIIRS BT differences consistently have values <0.08 K. In the case of M13, the Spectral Response Function of NOAA-21 is significantly different from that of NOAA-20 and S-NPP. As a result, the inter-VIIRS BT differences between NOAA-21 and NOAA-20/S-NPP have values of about 0.234 to 0.312 K, respectively. These BT differences are still comparable to the 0.2 K difference between VIIRS and MODIS.

We also discovered that substantial differences in SRFs between NOAA-21 and NOAA-20/S-NPP, particularly in M13, can lead to notably steep slopes, reaching about 0.035 K/K, in the changes of direct BT differences with SSTs rising. However, the double-difference method significantly reduces such slopes to below 0.0033 K/K after accounting for the SRF difference in CRTM, highlighting its effectiveness in mitigating the impact of SRF differences among different VIIRS instruments.

The M12 operates as a shortwave infrared channel. We observed distinct different performances between M12 daytime and nighttime data. For example, for S-NPP measurements, larger O-B ΔBTs with greater uncertainties and noticeable seasonal variations are observed in its daytime data. Conversely, larger drift and steeper slopes in the changes of O-B ΔBTs with increasing BTs are found in its nighttime data. Furthermore, M12 exhibits greater spatial variations in O-O ΔBTs during daytime compared to nighttime. All these features for M12 are closely linked to the influence of solar contribution through sea surface reflection on the TOA radiance during daytime.

In conclusion, our study demonstrates that the RTM-based TEB quality evaluation method is robust and versatile, assessing both long-term sensor stability and inter-sensor consistency. By employing CRTM simulation as the transfer reference, we provide a comprehensive quality evaluation of VIIRS TEB data, ensuring the stability of S-NPP VIIRS and consistency between S-NPP, NOAA-20, and NOAA-21. Furthermore, the double-difference approach mitigates uncertainties and biases inherent to CRTM simulations and further minimizes the influence of SRF differences, establishing a robust mechanism for assessing inter-sensor consistency. These findings are meaningful, as the stability and consistency of VIIRS TEBs, the focal points of this study, are crucial for maintaining and upholding the data quality of downstream VIIRS EDR products and advancing Earth science research and climate applications.

**Author Contributions:** Conceptualization, F.Z., X.S. and C.C.; methodology, F.Z., X.J., T.-C.L. and X.S.; software, F.Z., Y.C., X.J. and T.-C.L.; validation, F.Z., X.S., W.W. and Y.C.; formal analysis, F.Z., W.W. and X.S.; investigation, F.Z., X.S. and Y.C.; resources, C.C., X.S. and Y.C.; writing—original draft preparation, F.Z.; writing—review and editing, F.Z., X.S., C.C., Y.C., W.W., T.-C.L. and X.J.; visualization, F.Z.; supervision, C.C. and X.S.; project administration, C.C., W.W. and X.S.; funding acquisition, C.C. and X.S. All authors have read and agreed to the published version of the manuscript.

**Funding:** This study was supported by NOAA grant NA19NES4320002 (Cooperative Institute for Satellite Earth System Studies—CISESS) at the University of Maryland/ESSIC.

**Data Availability Statement:** ERA5 hourly data on single levels from 1940 to present: "https://cds.climate.copernicus.eu/cdsapp#!/dataset/reanalysis-era5-single-levels?tab=form (accessed on 1 April 2023)"; ERA5 hourly data on pressure levels from 1940 to present: "https://cds.climate.copernicus.eu/cdsapp#!/dataset/reanalysis-era5-pressure-levels?tab=form (accessed on 1 April 2023)"; NOAA STAR version 2 reprocessed S-NPP VIIRS data: "https://www.aev.class.noaa.gov/saa/products/search?sub_id=0&datatype_family=RPVIIRSSDR&submit.x=26&submit.y=12 (accessed on 1 April 2023)"; Operational VIIRS data for S-NPP/NOAA-21/NOAA-20: "https://www.aev.class.noaa.gov/saa/products/search?sub_id=0&datatype_family=VIIRS_SDR&submit.x=22&submit.y=6 (accessed on 1 April 2023)".

**Acknowledgments:** The scientific results and conclusions, as well as any views or opinions expressed herein, are those of the author(s) and do not necessarily reflect those of NOAA or the Department of Commerce.

**Conflicts of Interest:** The authors declare no conflict of interest.

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
