# Peer review of "Evaluation of VIIRS Thermal Emissive Bands Long-Term Calibration Stability and Inter-Sensor Consistency Using Radiative Transfer Modeling"

_remotesensing, doi:10.3390/rs16071271_

Round 1

Reviewer 1 Report

Comments and Suggestions for Authors

See attached Word document

Reviewer 2 Report

Comments and Suggestions for Authors

The manuscript evaluates the long-term accuracy and stability of the Visible Infrared Imaging Radiometer Suite (VIIRS) onboard Suomi National Polar-orbiting Partnership (S-NPP), NOAA-20, and NOAA-21 satellites. The assessment is conducted with thoroughness and attention to detail, ensuring the quality of the data product across various aspects. Additionally, the authors not only present potential limitations but also delve into the underlying causes, illustrated through plots of Spectral Response Function (SRF) and weight functions.

This study holds significant importance as it provides a comprehensive evaluation of the VIIRS data product spanning over 8 years. Such meticulous analysis instills confidence within the scientific community reliant on this data and its downstream products, particularly in climate change studies. I would recommend acceptance of this manuscript pending resolution of the following two issues:

1. In Figure 9, the trend line of NOAA-21 which is supposed to be in red color, is not visible, probably due to lack of color contrast against the background.

2. The double difference technique for spaceborne instrument evaluation has been used demonstrated as robust in previous studies. For example, Berg et al. (2016) have employed this method in comparing two instruments and Chen et al. (2017) applied it in a comparative analysis involving three instruments. Therefore, I suggest to include necessary references to support this methodology.

·      Berg, W., and Coauthors, 2016: Intercalibration of the GPM Microwave Radiometer Constellation. J. Atmos. Oceanic Technol., 33, 2639–2654, https://doi.org/10.1175/JTECH-D-16-0100.1.

and related references.

Reviewer 3 Report

Comments and Suggestions for Authors

 This paper presents a result of the long-term calibration stability examination for thermal infrared bands of S-NPP/VIIRS, and an inter-comparison case among three VIIRSs on NOAA-20 and NOAA-21 as well as S-NPP. Simulation results derived with a radiative transfer model, CRTM, are applied as the reference of the comparison, allowing to accumulate a large number of data for statistical analysis. These comparisons prove that the infrared bands of S-NPP/VIIRS are almost stable during a long period, February 2012 to August 2020, with revealing a very slight temporal drift. The differences in radiance among VIIRSs are also small and stable during about a half year.  This study gives a convincing evidence for the reliability of VIIRS in extracting signs of decadal climate changes from the observation time series.  

Major comments

 Calibration for sensors on satellite after launch is fundamental to perform correct measurements in a long period with eliminating the temporal degradation, but is difficult in principle. The method proposed in this study, comparing to simulations with a radiative transfer model, will be capable of being applied to estimating the sensor calibration stability. I therefore think that this paper is suitable to this journal, and will contribute to improve the remote sensing data quality. However, I think that there are crucial points to be confirmed for right understanding of the method (and to be explained in the text).

1. P7 L256, Does the "double difference (O-O) ∆BTs" correspond to "O-O" or "(O-B)-(O-B)"?

2. P14 L429, " the daily means of O-O ∆BTs between ~", Does the daily mean correspond to "(daily mean of O) - (daily mean of O), or daily mean of (O-O)? I think that the collocated data among three sensors should be required at least for the latter case: If it is right, describe briefly the procedure for gathering collocated data (e.g., the admissible spatial and temporal interval as collocation).

Minor comments and questions

3. P4 L124, Is F-factor time-dependent or not?

4. P8 L267, Describe also in the text the definition of the uncertainty (may be 1𝜎 of O-B during each month?).

5. P9 L286, "O-B differences during daytime ~", Are the sunglint areas included in evaluation data of M12 ?

Round 2

Reviewer 3 Report

Comments and Suggestions for Authors

I appreciate your response to my comments and questions. I have been satisfied by the responses and the corresponding modifications. I think that this paper is sufficient to be published.